# Investigating the risk factors for seroprevalence and the correlation between CD4+ T-cell count and humoral antibody responses to *Toxoplasma gondii* infection amongst HIV patients in the Bamenda Health District, Cameroon

**Eugene Enah Fang**[1], **Raymond Babila Nyasa**[1,2]*, **Emmanuel Menang Ndi**[2], **Denis Zofou**[2], **Tebit Emmanuel Kwenti**[3], **Edith Pafoule Lepezeu**[4], **Vincent P. K. Titanji**[2,5], **Roland N. Ndip**[1,6]

1 Department of Microbiology and Parasitology, University of Buea, Buea, Cameroon, 2 Biotechnology Unit, Faculty of Science, University of Buea, Buea, South West Region, Cameroon, 3 Department of Medical Laboratory Sciences, Faculty of Health Science, University of Buea, Buea, South West Region, Cameroon, 4 Bamenda Regional Hospital, Bamenda, North West Region, Cameroon, 5 Cameroon Christian University, Bali, North West Region, Cameroon, 6 Laboratory for Emerging Infectious Diseases, University of Buea, Buea, South West Region, Cameroon

* nyasab@yahoo.co.uk

## Abstract

### Background

Toxoplasmosis is caused by an obligate intracellular tissue protozoan parasite, *Toxoplasma gondii* that infect humans and other warm-blooded animals. Transmission to humans is by eating raw or inadequately cooked infected meat or through ingestion of oocysts that cats have passed in faeces. Studies have shown life-threatening and substantial neurologic damage in immunocompromised patients; however, 80% of humans remain asymptomatic. The aim of this study was to determine the seroprevalence of *Toxoplasma gondii* infection in HIV positive patients and the risk factors associated with the infection, and to investigate the correlation between CD4+ T-cell count and toxoplasma specific antibodies as possible predictors of each other amongst HIV patients in the Bamenda Health District of the North West Region of Cameroon.

### Methods

A cross-sectional study was conducted, in which 325 HIV patients were recruited for administration of questionnaire, serological diagnosis of *T. gondii* and measurement of CD4+ T-cell count. Bivariate and multivariate logistic regression was used to identify risk factors associated with *T. gondii* infection while the linear regression was used to investigate the relationship between CD4+ T-cell count and antibody levels against *T. gondii*.

**Data Availability Statement:** All relevant data are within the paper and its Supporting Information files.

**Funding:** The author(s) received no specific funding for this work.

**Competing interests:** The authors have declared that no competing interests exist.

**Abbreviations:** AIDS, Acquired Immune Deficiency Syndrome; ANOVA, Analysis of variance; ARV, Anti Retroviral; BHD, Bamenda Health District; CD4+, Cluster of Differentiation; EIA, enzyme immunosorbent assay; EDTA, Ethylenediaminetetraacetic Acid; ELISA, Enzyme Linked Immunosorbent Assay; FHS, Faculty of Health Sciences; HIV, Human Immune Deficiency Virus; IBM, International Business Machine; IgG, Immunoglobulin G; IgM, Immunoglobulin M; IRB, Institutional Review Board; OR, Odd Ratio; RDPH, Regional Delegation of Public Health; *T. gondii, Toxoplasmosis gondii*; WHO, World Health Organization.

## Results

The findings showed that, majority (45.8%) of HIV patients suffered from chronic (IgG antibody) infection, and 6.5% from acute (IgM and IgM/IgG antibody) toxoplasma infection. The overall sero-prevalence of *T. gondii* infection amongst HIV patients was 50.5%. On the whole, 43 men (45.7%) and 127 women (55%) presented with anti- *T. gondii* antibodies; however, there was no significant difference amongst males and females who were positive to *T. gondii* infection ($p = 0.131$). Marital status ($p = 0.0003$), contact with garden soil ($p = 0.0062$), and garden ownership ($p = 0.009$), were factors that showed significant association with *T. gondii* infection. There was no significant difference ($p = 0.909$) between the mean CD4+ T-cell count of HIV patients negative for toxoplasma infection (502.7 cells/mL), chronically infected with *T. gondii* (517.7 cells/mL) and acutely infected with *T. gondii* (513.1 cells/mL). CD4+ T-cell count was neither a predictor of IgM antibody titer ($r = 0.193$, $p = 0.401$), nor IgG antibody titer ($r = 0.149$, $p = 0.519$) amongst HIV patients acutely infected with *T. gondii*.

## Conclusion

The findings from this study underscore the need to implement preventive and control measures to fight against *T. gondii* infection amongst HIV patients in the Bamenda Health District.

## Introduction

Toxoplasmosis caused by the obligate intracellular parasite, *Toxoplasma gondii (T. gondii)* is a globally prevalent zoonotic disease of warm-blooded animals, including man. Within the human population, the infection is clinically silent in immunocompetent host and accounts for focal encephalitis, headache, confusion, motor weakness and fever amongst patients with acquired immunodeficiency syndrome (AIDS) or immuno-compromised patients [1]. In the absence of treatment, the disease progression results in seizures, stupor, coma and finally death [2]. The infection is contracted through consumption of oocyst in contaminated water, food or from inadequately cooked infected meat. Other means of infection include blood transfusion and organ transplantation [3].

Knowledge of *T. gondii* seroprevalence and its associated risk factors is important to predict the risk of infection in humans. Studies elsewhere have implicated consumption of raw garden produce, level of education and age as risk factors of the infection [4–6].

Furthermore, toxoplasmosis has been described as the commonest cerebral opportunistic infection in HIV infected patients [7] and a common cause of mortality amongst severely immunosuppressed HIV patients [8]. It is known that a CD4+ T-cell count of $\leq 200$cells/mm$^3$ accounts for reactivation of latent infection in HIV patients [9]. It is therefore rational to think that the magnitude of humoral immune response (IgG and IgM) is a function of CD4+ T cell count, since following *T. gondii* infection, a rise of specific acquired immunity, including humoral immunity accompanied by parasite clearance from the body [10]. If this is true, then high antibody titer in HIV infected toxoplasma patients should positively correlate with CD4 + T cell count, and consequently, either may serve as a marker of the other.

## Methods

### Study area and setting

The study was carried out in the Bamenda Health District (BHD), located on the High Western Plateau of Cameroon from August to October 2018. Bamenda is an urban setting, which is both the Administrative Headquarter of Mezam Division and the North West Region of Cameroon. The weather is warm and wet in the rainy season, which last from March to October. In the dry season (November to February), the mornings are generally very cold, and the afternoons are very hot compared to the rainy season with an average annual temperature of 19.3˚c. This cosmopolitan city is made up of mainly civil servants and business men and a majority of the population is partly involved in small scale farming. The BHD has a population of over 350,000 inhabitants. It comprises of 17 health areas, 14 public, and 4 faith-based health facilities (1 Baptist, 1 Presbyterian and 2 Catholics) with the Bamenda Regional Hospital (public) being the leading health facility. Amongst the 18 health facilities, there are only four HIV treatment centers (Bamenda Regional Hospital, Nkwen Sub-divisional Medicalised Health Center, Nkwen Baptist Hospital and Mezam Polyclinic) in the BHD.

### Study design

A cross sectional study was conducted in the Bamenda Health District where data was collected from August to October 2018. Participants were recruited by a convenient sampling technique at the HIV treatment centers of the Bamenda Regional Hospital and the Nkwen Sub-divisional Medicalised Health Center. The Bamenda Regional Hospital being the referral hospital and leading in terms of the HIV/AIDS patient population in the health district was purposively chosen for this study, while the Nkwen Sub-divisional Medicalized Health Center was randomly selected from the other three HIV treatment centers within the health district. Assuming a prevalence of 69.6% from a previous study elsewhere (Yaounde) in Cameroon [11], the sample size was calculated using the formula of Suresh and Chandrashekara [12]. A value of 323 was achieved which was rounded up to 325 during implementation. Three hundred and twenty-five consenting HIV patients 12 years old and above, who were registered at the treatment centers as residents in the BHD were enrolled into the study at the level of the treatment center during a visit. Participants were administered a structured questionnaire and 3 mL of venous blood was collected from each, into a dry tube for serological diagnosis of *T. gondii* infection, and another 3 mL collected into Ethylenediaminetetraacetic Acid (EDTA) tube for measurement of CD4+ T-cell count.

### Administration of questionnaires

Ten questionnaires were pretested at the treatment center of Ndop District Hospital in the North West region of Cameroon in June 2018. After the pretest exercise, the questionnaire was adapted to be completed within 7–10 minutes including keeping the question of age as an open-ended question. Validated questionnaires were administered to participants of all age groups to obtain data on demography (age of participants, gender, marital status, religion, level of education and occupation), risk factors and if participants had any clinical signs and symptoms of *T. gondii* infection, or a history of the infection. Risk factors investigated in this study were; duration on ARV, history of default to HIV/AIDS therapy, availability of cats in participant's home or neighborhood, garden ownership and nature of participant's contact with garden soil, consumption of raw and/or inadequately cooked meat and/or vegetables, source of domestic water supply, and treatment and storage condition of drinking water.

## Laboratory analysis

**Serological diagnosis of *T. gondii* infection.** Serological screening for Toxoplasma infection was achieved using Aria Toxo IgG/IgM Combo Rapid Test with accuracy of 94.9% and 97.8% IgG and IgM respectively (CTK Biotech, Inc., San Diego California, USA) according to the manufacturer's instructions [13]. The kit is a lateral flow chromatographic immunoassay; it detects simultaneously the presence of *T. gondii* IgG and IgM specific antibodies in patient's blood. Antibodies against *T. gondii* in patient's serum, binds to recombinant *T. gondii* antigen conjugated with colloidal gold and the coloured complex is captured in the test window coated with mouse anti-human antibody. In essence, 10 uL of serum obtained from patient's blood was applied to the sample pad alongside 70 uL of kit's diluent and the result was read after 10 minutes.

The amount of *T. gondii* IgM antibodies was estimated for IgM positive samples using the Biorex diagnostic *T. gondii* IgM (Antrim, United Kingdom) following manufacturer's instructions. The *T. gondii* IgM enzyme immunosorbent assay (EIA) Test kit is a solid phase EIA based on immunocapture of IgM antibodies in patient's serum. The presence of *T. gondii* specific antibodies in patient's serum complexes with added recombinant *T. gondii* antigen-enzyme conjugate and reveals a coloured product upon addition of substrate. In brief, following addition of 100 uL of negative control, cut-off calibrator and positive control in appropriate wells, 100 uL of specimen diluent and 5 uL of specimen was added to each of the other wells followed by incubation at 37˚C for 30 minutes and repeated washing. Except for the blank well, 100 uL of conjugate was added to each well and incubated at 37˚C for 30 minutes, followed by repeated washing; after which 50uL of substrate A and 50uL of substrate B were added, and microwell plate was incubated at 37˚C for 10 minutes. The reaction was stopped by addition of 50 uL of stop solution and the optical density was read at 450nm and 650 nm within 30 minutes, using the Emax Precision Microplate ELISA reader (Molecular Divices, California, USA).

The amount of IgG antibodies specific for *T. gondii* in the IgM positive samples was estimated using the Erbalisa® Toxoplasma IgG (Calbiotec, California, USA) according to manufacturer's instructions. The assay is composed of pre-coated wells with purified *T. gondii* antigen, to which patient's serum is added for *T. gondii* specific antibodies to bind; if present, an enzyme conjugate is added, which releases a coloured product upon addition of substrate. The assay was carried out by dispensing 100 uL of diluted sample, calibrator, controls and diluent or blank into appropriate wells, and incubated at room temperature for 20 minutes, before washing three times. This was followed by addition of enzyme conjugate and incubation was done at room temperature for 20 minutes before washing. One hundred microlitres of TMB substrate was added to each well and incubated at room temperature for 10 minutes before the reaction was stopped using 100 uL of stop solution. The optical density was read at 450 nm and 650 nm using Emax Precision Microplate ELISA reader (Molecular Divices, California, USA).

## Measurement of CD4+ T-cell count

The Pima™ Analyser was used (SN; PIMA-A-005920; Alere Technology Gmbh Loebstedter Street 103–105 Jena, Germany), in the measurement of CD4+ T-cell count, which employs a static image analysis. In essence plasma obtained from whole blood collected in EDTA tube was kept at 18˚C and analysis was done within 24hrs. Plasma sample was loaded to PIMA CD4 + test cartridge and ran in the analyser following insertion of test cartridge. Results were automatically calculated and absolute T-cell counts displayed.

## Ethical considerations

Ethical clearance for this study was obtained from the University of Buea, Faculty of Health Science (FHS) Institutional Review Board (IRB), reference number 2018/2244UB/SG/IRB/FHS and the Bamenda Regional Hospital Institutional Review Board, reference number 26/APP/RDPH/RHB/IRB. Administrative authorization was obtained from the Regional Delegation of Public Health for the North West Region (RDPH), reference number 233/ATT/NWR/RDPH. Only consenting participants from the study population were recruited into the study, and they all signed the informed consent form. For participants below the age of 21 years old, a signed ascent form was also obtained from their parents or guardians to enable them take part in the research. The participant's selection exercise was completely a random process.

## Statistical analysis

The data were analyzed using Microsoft excel 2010 and *IBM SPSS* version 25 with significance level judged at *p<0.05*. Descriptive statistics was computed to establish the frequency of each response type in the sociodemographic characteristic and the sero-prevalence in the study population. Independent variables (socioeconomic, demographic and nutritional risk factors) were analyzed using bivariate and multivariate logistic regression to eliminate confounders and ascertain the level of significance as suspected risk factors of *T. gondii* infection. A one-way ANOVA test was conducted to compare the difference between the mean CD4+ T cell count groups of HIV/AIDS patients who were either *T. gondii* infection negative, chronic toxoplasma infection or acute toxoplasma infection. Linear regression analysis was computed to establish relationships between IgM and IgG antibody levels against *T. gondii* with CD4+ T cell count.

## Results

The overall seroprevalence of *T. gondii* in the BHD of the North West Region of Cameroon was 50.5% (*CI* = 45.3%-55.7%). The prevalence of chronic (IgG antibody) toxoplasma infection was 45.8% and acute (IgM antibody) infection was 6.5%. The ages of the study participants ranged from 12 to 73 years with an overall mean age of 41.83±13.41years. Majority of the participants, 56.9% (185) were above the age of 41 years, while 43.1% (140) were below the age of 41 years. A Pearson Chi-Square test indicated that there was a significant difference in the seroprevalence of *T. gondii* infection across the age groups of HIV patient $\chi^2$ (2) = 8.180, *p* = 0.017, with older patients having a higher prevalence (Table 1).

Majority of HIV patients recruited for the study were females 71.1% (231), as compared to males 28.9% (94). One hundred and twenty-seven (55%) females as compared to 43 (45.7%) males were positive for *T. gondii* infection. A Pearson Chi-Square test analysis indicated that there was no statistically significant difference between the number of male and female HIV patients who were positive for *T. gondii* infection $\chi2$ (1) = 2.283, *p* = 0.131.

## Risk factors of *T. gondii* infection in the study population

In the bivariate and multivariate logistic regression analysis, factors that were statistically significantly associated to the development of *T. gondii* in HIV patients included demographic factors; marital status (*p≤0.001*) and socioeconomic factors; contact with garden soil (*p* = 0.006), and garden ownership (*p* = 0.009).

**Table 1. Prevalence of *T. gondii* by age, antibodies, and gender in the BHD.**

| Characteristics | Positive No (%). | Negative No (%) | Total No (%) | χ2 | *p-value* |
|---|---|---|---|---|---|
| **Antibodies** | | | | | |
| **IgG only** | 149(45.8) | 176(54.2) | 325(100) | - | - |
| **IgM only** | 6(1.8) | 319(98.2) | 325(100) | - | - |
| **IgG/IgM** | 15(4.6) | 310(95.4) | 325(100) | - | - |
| **Gender** | | | | | |
| **Female** | 127(55.0) | 104(45.0) | 231(100) | 2.283 | 0.131 |
| **Male** | 43(45.7) | 51(54.3) | 94(100) | - | - |
| **Age** | | | | | |
| **≤21** | 10 (3.1) | 20 (6.2) | 30(9.3) | 8.180 | 0.017 |
| **22–40** | 52(16) | 58 (17.8) | 110(33.8) | - | - |
| **41≥** | 108 (33.2) | 77 (23.7) | 185(56.9) | - | - |

% = percentage, χ2 = chi square.

## Demographic, nutritional and socio-economic factors associated with *T. gondii* infection

*T. gondii* infection was significantly more prevalent among HIV patients who were Single/Divorced/widow(er) 54.5% (177/325) (Table 2), compared to HIV patients who were married 45.5% (148/325). The risk of developing Toxoplasma infection was about 2.4 times significantly higher among HIV positive patients who were either Single/Divorced/widow(er) ($OR = 2.41$, $CI = 1.49$–$3.88$, $p \leq 0.001$), compared to patients who were married.

In the bivariate logistic regression, no nutritional factor (consumption of improperly cooked or raw meat $p = 0.798$ and consumption of improperly cooked vegetable or salad $p = 0.124$) was significantly associated to the development of Toxoplasma infection (Table 2).

The proportion of HIV patients who presented with Toxoplasma infection (Table 2), and had been in contact with garden soil 74.2% (241/325) was significantly higher compared to HIV patients who presented with Toxoplasma infection without contact with garden soil 25.8% (84/325), ($p = 0.006$). The risk of developing toxoplasmosis was found to be about 7 times significantly higher among HIV patients who had been in contact with garden soil ($OR = 7.03$, $CI = 1.74$–$28.79$, $p = 0.006$), compared to HIV patients who had no contact with garden soil (Table 2).

Majority of Toxoplasma infected HIV patients owned gardens 75.7% (246/325), compared to those who did not own a garden 24.3% (79/325), ($p = 0.009$). The risk of developing Toxoplasma infection was found to be about 2 times significantly higher among HIV patients who owned gardens ($OR = 2.19$, $CI = 1.21$–$3.96$, $p = 0.009$), compared to HIV patients who did not own a garden (Table 2).

## Determination of CD4+ T-cell count as a predictor of acute Toxoplasma infection

One-way analysis of variance (ANOVA) test indicated that there was no significant difference ($p = 0.909$) between the mean CD4+ T-cell count of HIV patients who were negative for toxoplasmosis and those who were chronically infected or acutely infected with *T. gondii* infection (Fig 1). Toxoplasma infection was not significantly associated with CD4+ T-cell counts ≤200cells/mL (*p = 0.536*).

**Table 2. Association of *T. gondii* infection and demographic, nutritional and socio-economic factors among HIV patients in the BHD.**

| Variable | Seroprevalence | Risk factors for *T. gondii* seroprevalence | | | |
| --- | --- | --- | --- | --- | --- |
| | No (%) | Bivariate Analysis | | Multivariate Analysis | |
| | | COR (95%CI) | p-value | AOR (95%CI) | p-value |
| **Marital status** | | | | | |
| Married | 148 (45.5) | 1 | | | |
| Single/Divorce/widow(er) | 177 (54.5) | 1.81 (1.17–2.83) | 0.008 | 2.41(1.49–3.88) | ≤0.001 |
| **Age** | | | | | |
| ≤21 | 30 (9.2) | | | | |
| **22–40** | 110 (33.8) | 1.79(0.77–4.18) | 0.176 | | |
| ≥41 | 185 (57) | 2.74(1.22–6.19) | 0.015 | | |
| **Occupation** | | | | | |
| Business | 87 (26.8) | 1 | | | |
| Farming | 93 (28.6) | 1.24(0.69–2.23) | 0.479 | | |
| Others | 117 (36) | 1.05(0.60–1.83) | 0.858 | | |
| Student | 28 (8.6) | 0.44(0.18–1.09) | 0.075 | | |
| **Consumption of raw/poorly cooked meat** | | | | | |
| No idea | 85 (26.1) | 1 | | | |
| No | 189 (58.2) | 1.07 (0.6405–1.7853) | 0.798 | | |
| Yes | 51 (15.7) | 0.77 (0.3815–1.5366) | 0.453 | | |
| **Consumption of raw/poorly cooked vegetable** | | | | | |
| No | 67 (20.6) | 1 | | | |
| Yes | 258 (79.4) | 1.53 (0.8904–2.6310) | 0.124 | | |
| **Contact with garden soil** | | | | | |
| No | 84(25.8) | 1 | | | |
| Yes | 241(74.2) | 3.44(2.02–5.87) | <0.001 | 7.03(1.74–28.79) | 0.006 |
| **Owns a garden** | | | | | |
| No | 79 (24.3) | 1 | | | |
| Yes | 246 (75.7) | 2.63(1.55–4.46) | ≤0.001 | 2.19(1.21–3.96) | 0.009 |
| **Owns a cat** | | | | | |
| No | 218 (67.1) | 1 | | | |
| Yes | 107 (32.9) | 1.60(1.00–2.56) | 0.049 | | |
| **Neighbour owns a cat** | | | | | |
| No | 71 (21.2) | 1 | | | |
| Yes | 254 (78.2) | 2.07(1.20–3.54) | 0.008 | | |

COR = Crude odd ratio; AOR = Adjusted odd ratio; % = percentage; CI = Confidence Interval.

## Relationship between IgM antibodies against Toxoplasma infection and CD4+ T-cell count in HIV patients with *T. gondii* infection

Amongst the 21 patients with acute infection (positive for IgM antibodies to *T. gondii*, IgM antibody titers decreased with increasing CD4+ T-cell counts ($r = 0.193$), but there was no statistically significant difference ($p = 0.401$) in IgM antibody titer at different levels of CD4+ T-cell counts (Fig 2).

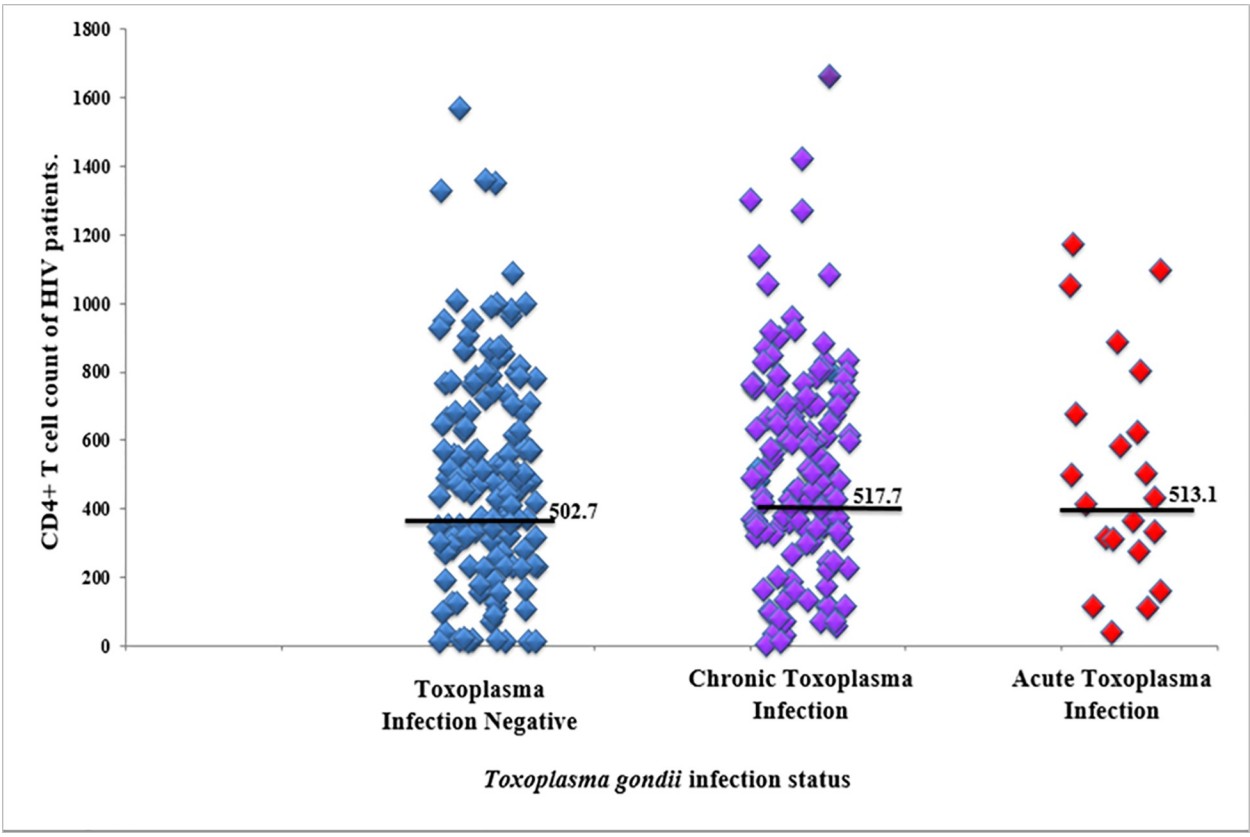

**Fig 1. Mean CD4+ T-cell counts between different status of Toxoplasma infection.** Toxoplasma negative = participants with neither IgM nor IgG antibodies against T. gondii infection. Chronic Toxoplasma Infection = participants who were positive only for IgG antibodies against T. gondii. Acute Toxoplasma Infection = participants who were at least positive for IgM antibodies against T. gondii infection.

## Relationship between IgG antibodies against Toxoplasma infection and CD4+ T cell count in HIV patients with acute toxoplasmosis

Amongst the 21 HIV patients acutely infected with *T. gondii*, there was an increase in IgG antibody titers as the amount of CD4+ T cell counts increased, but there was no linear relationship ($p = 0.519$) as seen in Fig 3.

## Discussion

Results from this study revealed a Toxoplasma infection prevalence of 50.5% (164/325). This is less than the prevalence reported in other studies where the prevalence of latent toxoplasmosis was 93.3% in Addis Ababa, Ethiopia, [14]; 60% in Muttu Karl Hospital, Gonder, Ethiopia [15]; 96.3% in North Iran [16] and 69.9% in HIV patients attending the University Teaching Hospital in Yaoundé, Cameroon [11]. The high prevalence of chronic Toxoplasma infection (45.8%) relative to acute infection (6.5%) can be explained by the improved HIV treatment package which reduces mortality and the duration of immune suppression in HIV patients, thus prolonging life expectancy and increasing the number of Toxoplasma chronically infected patients.

Toxoplasma infection was found to be significantly associated ($p = 0.006$) with HIV patients who had contact with garden soil (74.2%) and those who owned gardens (75.7%, $p = 0.009$). However, the infection was not significantly associated with consumption of raw or

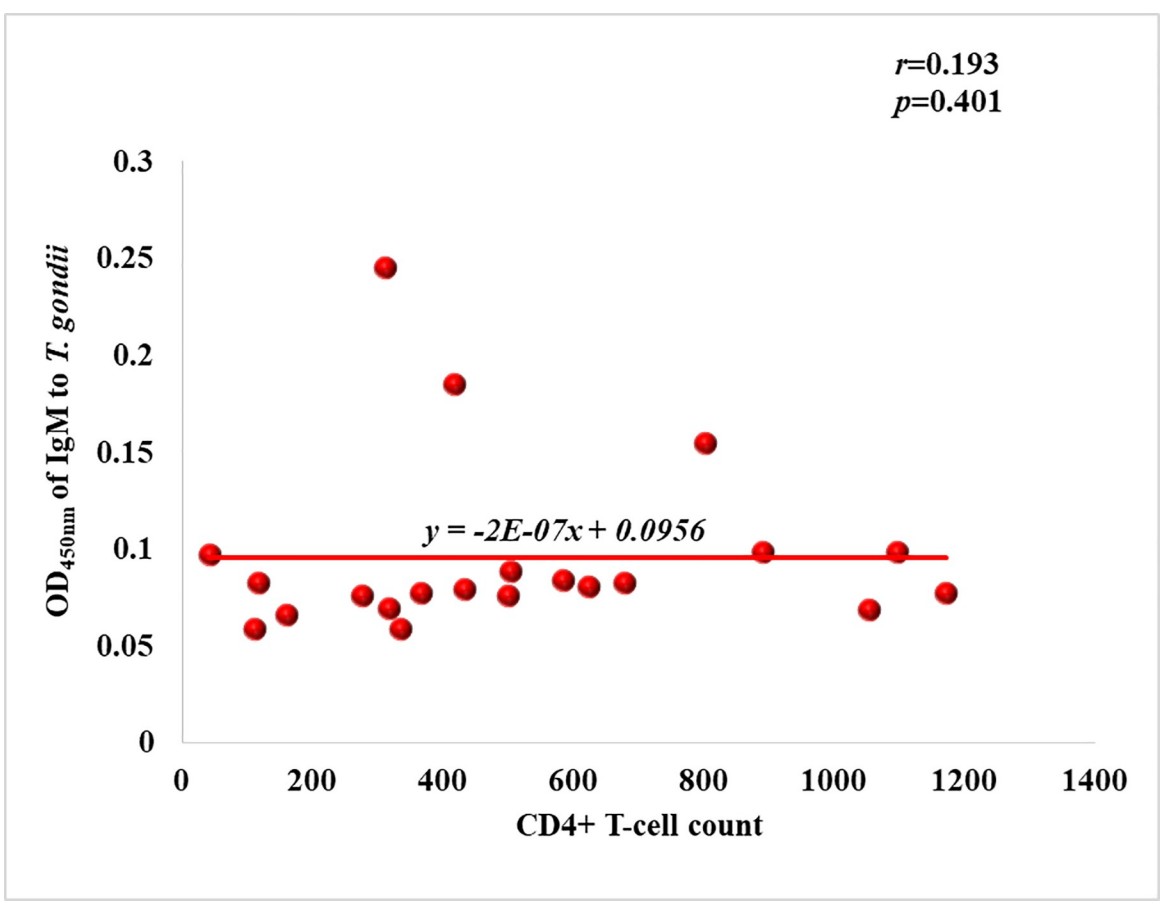

**Fig 2. IgM antibodies against Toxoplasma infection and CD4+ T cell count in HIV patients.**

inadequately cooked meat ($p = 0.45$), and consumption of raw or inadequately cooked vegetable ($p = 0.124$). This finding is contrary to what was obtained in Ethiopia amongst a similar group of patients, where anti-*T. gondii* seropositivity was significantly associated with raw meat consumption ($p = 0.025$) [17]. Consumption of raw meat is not a common practice in the BHD as most of the staple dishes in this locality entail rigorous cooking. In this study; age, cat ownership or neighbors owning cat, were not significantly associated with Toxoplasma infection following multivariate analysis. This is contrary to the observation amongst HIV infected women in Ethiopia where the age group 28 to 37 years was a significant risk factor for Toxoplasma infection [18]. In the same light, studies carried out in Lagos, Nigeria showed that Toxoplasma infection was associated with participants living in close proximity with cats ($p = 0.013$), contradicting the findings from this study [19]. This study also showed that Toxoplasma infection was significantly associated with single/divorce/widow(er) ($p \leq 0.001$), which is contrary to observations in Southern Iran where marital status was not a risk factor for the disease [20]. These findings suggest that the epidemiology of *T. gondii* infection may vary in different settings.

Although other studies [21, 22], have shown that CD4+ T cell count threshold of $\leq$200cells/mL was indicative of onset of clinical manifestations of toxoplasma infection, in this study, it was observed that there was no significant difference between the mean CD4+ T-cell count of non-infected, chronically infected and acutely infected patients of toxoplasma infection ($p = 0.909$). The fact that there was no significant correlation between CD4+ T-cell count and

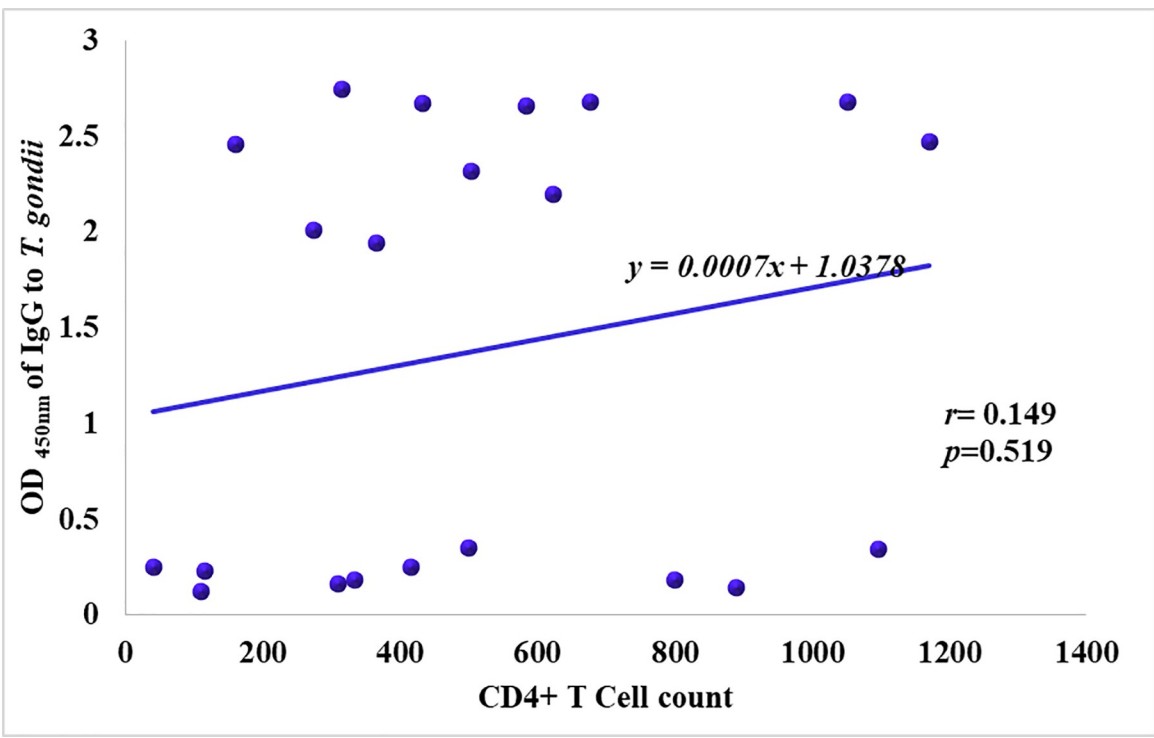

**Fig 3. IgG antibodies against Toxoplasma infection and CD4+ T cell count in HIV patients.**

IgM antibody titer ($r$ = 0.193, $p$ = 0.401) suggests that the quantity of IgM antibodies elicited in acutely infected patients is independent of the level of CD4+ T-cells present, although a study in South Brazil reported that CD4+ T-cell count $\leq$350cells/mm$^3$ of blood, was significantly associated ($p$ = 0.001) with development of acute toxoplasmosis [23]. Antibody titers of IgG showed a positive correlation with increasing CD4+ T-cells; however, it was not significant ($r$ = 0.149, $p$ = 0.519). This suggest that CD4+ T-cell count does not differ from one serological status to the other in *T. gondii* infection. This suggests that IgG antibody titer cannot be used to predict CD4+ T-cell count in *Toxoplasma gondii* patients presenting with IgG and IgM antibodies.

## Conclusion

The prevalence of *T. gondii* is still high, despite the successes recorded by the package put in place by WHO through the health system to control and manage opportunistic infections among HIV/AIDS patients. These findings therefore have profound clinical and epidemiological significance and call for more urgent action.

## Supporting information

**S1 Data. Raw data for toxo project.**
(XLSX)

## Acknowledgments

We wish to thank the staff of the Bamenda Regional Hospital HIV/AIDS treatment center for their collaboration and the Biotechnology Unit, Faculty of Science, University of Buea for helpful ELISA laboratory analysis.

## Author Contributions

**Conceptualization:** Eugene Enah Fang, Raymond Babila Nyasa, Vincent P. K. Titanji, Roland N. Ndip.

**Data curation:** Eugene Enah Fang, Raymond Babila Nyasa, Emmanuel Menang Ndi, Denis Zofou, Tebit Emmanuel Kwenti, Edith Pafoule Lepezeu.

**Formal analysis:** Eugene Enah Fang, Raymond Babila Nyasa, Emmanuel Menang Ndi, Edith Pafoule Lepezeu.

**Funding acquisition:** Eugene Enah Fang, Raymond Babila Nyasa, Denis Zofou, Edith Pafoule Lepezeu.

**Investigation:** Eugene Enah Fang, Raymond Babila Nyasa, Emmanuel Menang Ndi, Denis Zofou, Tebit Emmanuel Kwenti, Edith Pafoule Lepezeu, Roland N. Ndip.

**Methodology:** Eugene Enah Fang, Raymond Babila Nyasa, Emmanuel Menang Ndi, Denis Zofou, Tebit Emmanuel Kwenti, Edith Pafoule Lepezeu, Vincent P. K. Titanji, Roland N. Ndip.

**Project administration:** Eugene Enah Fang, Raymond Babila Nyasa, Denis Zofou, Tebit Emmanuel Kwenti, Vincent P. K. Titanji, Roland N. Ndip.

**Resources:** Eugene Enah Fang, Raymond Babila Nyasa, Emmanuel Menang Ndi, Denis Zofou, Edith Pafoule Lepezeu, Roland N. Ndip.

**Software:** Eugene Enah Fang, Raymond Babila Nyasa.

**Supervision:** Eugene Enah Fang, Raymond Babila Nyasa, Emmanuel Menang Ndi, Denis Zofou, Tebit Emmanuel Kwenti, Edith Pafoule Lepezeu, Vincent P. K. Titanji, Roland N. Ndip.

**Validation:** Eugene Enah Fang, Raymond Babila Nyasa, Denis Zofou, Tebit Emmanuel Kwenti, Edith Pafoule Lepezeu, Vincent P. K. Titanji, Roland N. Ndip.

**Visualization:** Eugene Enah Fang, Raymond Babila Nyasa, Emmanuel Menang Ndi, Denis Zofou, Tebit Emmanuel Kwenti, Edith Pafoule Lepezeu, Vincent P. K. Titanji, Roland N. Ndip.

**Writing – original draft:** Eugene Enah Fang, Raymond Babila Nyasa, Roland N. Ndip.

**Writing – review & editing:** Eugene Enah Fang, Raymond Babila Nyasa, Emmanuel Menang Ndi, Denis Zofou, Tebit Emmanuel Kwenti, Edith Pafoule Lepezeu, Vincent P. K. Titanji, Roland N. Ndip.

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
