## [Decision Letter · Decision Letter 0]

15 Jun 2020

PONE-D-20-04369

Risk factors for sero-prevalence of Toxoplasma infection and the role of CD4+ T-cell count as a determinant of acute infection amongst HIV patients in the Bamenda Health District, Cameroon.

PLOS ONE

Dear Dr. Enah,

Thank you for submitting your manuscript to PLOS ONE. After careful consideration, we feel that it has merit but does not fully meet PLOS ONE’s publication criteria as it currently stands. Therefore, we invite you to submit a revised version of the manuscript that addresses the points raised during the review process.

There are several major problems in the understanding of this study, the way the statistical analysis has been performed and presented, as well in the interpretation of the findings. I urge you to carefully consider the best course of action for you with regards to this manuscript. If you feel it is possible to undertake a re-analysis and re-write of the manuscript in the required time frame, ensuring all of the areas of concern raised by all Reviewers have been thoroughly addressed, a revised manuscript will be reviewed and considered. If you do not feel it is possible to address all of these issues thoroughly, you may wish to consider taking the time necessary to do this and ensure the manuscript is technically sound and appropriately interpreted, before re-submitting here or elsewhere as a new manuscript.

We look forward to receiving your revised manuscript.

Kind regards,

Leanne Joy Robinson

Academic Editor

PLOS ONE

Journal Requirements:

2. Please include additional information regarding the survey or questionnaire used in the study and ensure that you have provided sufficient details that others could replicate the analyses. For instance, if you developed a questionnaire as part of this study and it is not under a copyright more restrictive than CC-BY, please include a copy, in both the original language and English, as Supporting Information. Moreover, please include more details on how the questionnaire was pre-tested, and whether it was validated.

3. In your Methods section, please provide additional information about the participant recruitment method and the demographic details of your participants. Please ensure you have provided sufficient details to replicate the analyses such as: a) the recruitment date range (month and year), b) a description of any inclusion/exclusion criteria that were applied to participant recruitment, c) a table of relevant demographic details, d) a statement as to whether your sample can be considered representative of a larger population, e) a description of how participants were recruited, and f) descriptions of where participants were recruited and where the research took place.

Additional Editor Comments (if provided):

The manuscript does not currently meet the requirements for publication in PLoS One. There are several major problems in the understanding of this study, the way the statistical analysis has been performed and presented, as well in the interpretation of the findings. The authors need to consider the best course of action. If it is possible to undertake a full re-analysis and re-write of the manuscript in the required time frame, ensuring all of the areas of concern have been thoroughly addressed, a revised manuscript will be reviewed and considered. If all of the above requirements cannot be thoroughly addressed, the authors may wish to consider taking the time to do this and ensure their manuscript is technically sound and appropriately interpreted, before re-submitting here or elsewhere as a new manuscript.

Reviewers' comments:

Reviewer's Responses to Questions

**Comments to the Author**

1. Is the manuscript technically sound, and do the data support the conclusions?

Reviewer #1: Partly

Reviewer #2: No

Reviewer #3: Yes

2. Has the statistical analysis been performed appropriately and rigorously? 

Reviewer #1: No

Reviewer #2: Yes

Reviewer #3: Yes

3. Have the authors made all data underlying the findings in their manuscript fully available?

Reviewer #1: Yes

Reviewer #2: Yes

Reviewer #3: Yes

4. Is the manuscript presented in an intelligible fashion and written in standard English?

Reviewer #1: Yes

Reviewer #2: Yes

Reviewer #3: Yes

5. Review Comments to the Author

Reviewer #1: The manuscript titled "Risk factors for sero-prevalence of Toxoplasma infection and the role of CD4+ T-cellcount as a determinant of acute infection amongst HIV patients in the Bamenda Health

District, Cameroon" was well written and contains good information. But it could not present the results in a tight manner. For example, the demographic, economic, and nutritional variables are related to each other and affected each other and should have been examined together. And it was necessary to determine the basis on which the variable is included for the multivariate examination.

Also, with respect to the numerical variables, the authors did not reveal the nature of the distribution of those variables and whether they followed the normal distribution in order to use Anova test and why not use T-Test?.

In addition, the authors treat the Odd ratio test on the basis that it quantites the association between  the variables and this is incorrect.  Odd ratio quantifies the strength of the association between two events. There are other tests to do this, such as a risk ratio (RR). On the other hand, we cannot say that the risk increases by 2.4 (table 2 etc..), in fact, the increase is only 1.4 because one is not calculated, as it means that there is no difference.

Other minor corrections include:

It would be better for the author to indicate the average temperatures in the study area. For the phrase "Toxoplama infection" throughout the manuscript, the  the genus Toxoplasma should be written in Italic. The age group 40 and more should be corrected (Table 1).

Reviewer #2: This paper analyzed the seroprevalence of toxoplasmosis in HIV-infected patients in the DHB, Cameroon. Then the authors searched for risk factors for toxoplasmosis. The technical part is well described and the paper is well written. However, it suffers from insufficiencies and inadequate data analysis, which flaws the conclusion.

There are several major problems in the understanding of this study. First of all, the level of CD4+ T cells is well-known as a determinant factor favoring Toxoplasma reactivation, it is not “hypothesized” (line 72). Why do the authors think that “it is necessary to determine the relationship between CD4+ T-cell count and antibody titer to current Toxoplasma infection” (line 75-76) to guide patient management? After primary infection, the residual level of specific IgG is very different from one patient to another, whatever their HIV status. This seems the announced goal of the study (which does not make real sense), but despite that, the main part of the paper is dedicated to antibody screening and risk factor analysis.

Regarding the serological analysis, the most faithful seroprevalence should be calculated on the detection of IgG, regardless the presence of IgM. Indeed, the sole presence of IgM does not necessarily means that it is an acute infection, as non-specific IgM can be detected with many assays. As it is a cross-sectional study, we don’t have the result of a serological follow-up confirming primary-acquired infection by the rise of specific IgG. Thus seroprevalence should be 164/325. Besides, it is incorrect to state that IgM-positive patients have “acute” infection (line 185), as IgM can persist for months or years after primary infection (or be non-specific). Only IgG avidity testing would help confirming possible recent infection. What is really missing here is the context of patient inclusion. Where they consulting for routine checkup and delivery of antiretroviral therapy, or because of clinical signs? This would make a difference in the interpretation of serological results. Anyway, as expected, the seroprevalence increases with age, this is no novelty. There is no need to re-analyze age categories in Table 2. Tables 2, 3 and 4 should be merged.

The analysis of CD4+ T cell counts among Toxoplasma-seropositive and Toxoplasma-seronegative patients does not make sense. Do the authors expect that it influences the contamination with the parasite, which is environmental and foodborne ? it only influences the reactivation of past infection, which is not addressed here. Not surprisingly, there is no correlation between IgG or IgM titers and CD4+ cell counts. Comments about a trend (lines 248-51 and 255-57) are overstated and should be removed.

In the Discussion, it would be interesting to focus on prevalence data obtained in other African countries on the same patient population (HIV+), and to search for possible explanations for this discrepancy with previous results from Cameroon: is the DHB particular in terms of demography, climate, cultural behavior…? This part is well discussed, but the setting of the studies used for comparison should be more precisely addressed (HIV? Pregnant women? Randomly selected population?).

The sentence: “Contrary to other studies, which have shown that CD4+ T cell count threshold of ≤200cells/mL was indicative of onset of clinical manifestations of toxoplasma infection, in this study, it was observed that there was no significant difference between the mean CD4+ T-cell count of non-infected, chronically infected and acutely infected patients of toxoplasma infection” demonstrates that the authors poorly understand the pathophysiology of the disease and mix up serological and clinical findings. Again, the population study (not described here) is probably not the same as the studies cited. Most patients included, I guess, had no clinical signs and visited the health care facility for routine checkup, not because of clinical signs. Additionally, the authors considered that the diagnosis relied on their serological results (i.e. IgG+ = chronic infection, IgM+= acute infection), which is perfectly wrong. Indeed, acute toxoplasmosis in HIV+ patients mostly results from reactivation, thus all patients with clinical signs AND specific IgG (NOT IgM) should considered at risk for reactivation.

Altogether, this paper relies mainly on a seroprevalence study and associated risk factors and should only present the related results.

Reviewer #3: This study has been done to evaluate the seroprevalence of T. gondii infection in 325 HIV positive patients and the risk factors associated with the infection. Authors investigated the threshold of CD4+ T- cell count associated with acute T. gondii infection in the Bamenda Health District of the North West Region of Cameroon.

Comments to authors:

Introduction

Line 58: please write the full scientific name, Toxoplasma gondii, when you mention the name for the first time in the manuscript

Lines 67-69: it would be useful to present data regarding the seroprevalence of T. gondii from different areas of the world, to better put this study in the big picture, and to make this study more interesting to international readers.

Line 69: reference 4 is a poster presentation, please provide full length papers as references

Line 71: reference 7 is a poster presentation of a case report. Authors should provide full length papers as references

Lines 99-100: authors to rephrase the sentence, it is not clear

Lines 120-153: Methods, Serological diagnosis of T. gondii infection, sensitivity and specificity would be good to provide for the serological tests

Lines 120, 128, 130, 179, 198, 203, 204, 206, 212, 216, 219 aso in the paper: “Toxoplasma infection”, please use full scientific name in scientific writing, “T. gondii infection”

Line 200: please write toxoplasmosis, with lower case t

Lines 320-321: the journal is not BMC Veterinary Research, please check and provide journal’s name

Line 377, list of abbreviation: “Toxoplasmosis gondii”, please write “Toxoplasma gondii”

6. PLOS authors have the option to publish the peer review history of their article (what does this mean?). If published, this will include your full peer review and any attached files.

Reviewer #1: No

Reviewer #2: No

Reviewer #3: No

---

## [Author Response · Author response to Decision Letter 0]

11 Jan 2021

Response to Reviewers.

Dear reviewers kindly fine in this later the response to the questions posted.

The topic of the manuscripts has been modified to “Investigating the risk factors for seroprevalence and the correlation between CD4+ T-cell count and humoral antibody responses to Toxoplasma gondii infection amongst HIV patients in the Bamenda Health District, Cameroon” to tie the research with the scientific outcome. the map have been removed because it does not significantly add value to the article. Information about the pretest procedure of the question is as described in line 124 in the questionnaire administration sub heading.

During analysis, the numeric variables followed an abnormal distribution and the analysis of variance test was used. 

Determining the relationship between CD4+ T-cell count to antibody titer was to evaluate the effect of the parasite on the immune system in the study population. The investigator captured the aspect of risk factors to ascertain the epidemiology situation of the parasite in the study population. The investigator used a highly specific and sensitive screening test to ascertain the status of infection of the parasite among the study population. It is also true that, the presence of IgM antibodies is the indication of onset of infection and/or recontamination. The inclusion criteria have been addressed in the revised manuscript line 119 indicating the patients were for their routine drug pick up. 

The analysis of CD4+T-cell counts among toxoplasma negative and positive patients was to proof the hypothesis that say immune depression (Low CD4+ T-cell count) futher puts the patients in danger to Toxoplasmosis complications.

---

## [Decision Letter · Decision Letter 1]

24 Mar 2021

PONE-D-20-04369R1

Investigating the risk factors for seroprevalence and the correlation between CD4+ T-cell count and humoral antibody responses to Toxoplasma gondii infection amongst HIV patients in the Bamenda Health District, Cameroon.

PLOS ONE

Dear Dr. Enah,

Thank you for submitting your manuscript to PLOS ONE. After careful consideration, we feel that it has merit but does not fully meet PLOS ONE’s publication criteria as it currently stands. Therefore, we invite you to submit a revised version of the manuscript that addresses the points raised during the review process.

The authors write the article in a good way. However, they need to address some concerns related to the statistical analysis as requested by reviewer 1. Also, they need to enhance the discussion part. The results contradict most of the previously published articles. Thus, the authors need to discuss the results critically. There are other minor corrections required to be addressed to enhance the quality of the article.

We look forward to receiving your revised manuscript.

Kind regards,

Mohammed Abdelfatah Mosa Alhoot, PhD

Academic Editor

PLOS ONE

Additional Editor Comments (if provided):

Authors write the article in good way but need to address the some concerns related to the statistical analysis as requested by the reviewer 1. Also they need to enhance the discussion part. The results come in contradiction with most of previously published articles and the author need to give explanation for the results. there are another minor correction required to improve the quality of the article.

Reviewers' comments:

Reviewer's Responses to Questions

**Comments to the Author**

1. If the authors have adequately addressed your comments raised in a previous round of review and you feel that this manuscript is now acceptable for publication, you may indicate that here to bypass the “Comments to the Author” section, enter your conflict of interest statement in the “Confidential to Editor” section, and submit your "Accept" recommendation.

Reviewer #1: (No Response)

Reviewer #3: (No Response)

Reviewer #4: (No Response)

2. Is the manuscript technically sound, and do the data support the conclusions?

Reviewer #1: Partly

Reviewer #3: Yes

Reviewer #4: Yes

3. Has the statistical analysis been performed appropriately and rigorously? 

Reviewer #1: No

Reviewer #3: Yes

Reviewer #4: Yes

4. Have the authors made all data underlying the findings in their manuscript fully available?

Reviewer #1: Yes

Reviewer #3: Yes

Reviewer #4: Yes

5. Is the manuscript presented in an intelligible fashion and written in standard English?

Reviewer #1: Yes

Reviewer #3: Yes

Reviewer #4: Yes

6. Review Comments to the Author

Reviewer #1: The manuscript titled "Investigating the risk factors for seroprevalence and the correlation between CD4+ T-cell count and humoral antibody responses to Toxoplasma gondii infection amongst HIV patients in the Bamenda Health District, Cameroon" was well written and contains good information. However, Demographic, socioeconomic and nutritional variables have a relationship with each other and affect each other, and they should be included in the multivariate statistical analysis together.

And it was necessary to determine the basis on which the variable is included for the multivariate examination.

In addition, the authors treat the Odd ratio test on the basis that it quantites the association between the variables and this is incorrect. Odd ratio quantifies the strength of the association between two events. There are other tests to do this, such as a risk ratio (RR). On the other hand, we cannot say that the risk increases by 2.4 (table 2 etc..), in fact, the increase is only 1.4 because one is not calculated, as it means that there is no difference.

Other minor corrections include:

It would be better for the author to indicate the average temperatures in the study area. For the phrase "Toxoplama infection" throughout the manuscript, the the genus Toxoplasma should be written in Italic. The age group 40 and more should be corrected (Table 1). Use three decimal digits for P value results.

Reviewer #3: Lines 67-69: Authors to rephrase the paragraph and provide references regarding the seroprevalence and risk factors associated with T. gondii infection among individuals with human immunodeficiency virus (HIV) from different areas of the world.

Lines 326-327: please delete BMC, the journal is Parasite Vectors

Reviewer #4: please check the uploaded file for corrections.

For the discussion need to find researches to support your findings

7. PLOS authors have the option to publish the peer review history of their article (what does this mean?). If published, this will include your full peer review and any attached files.

Reviewer #1: No

Reviewer #3: No

Reviewer #4: No

---

## [Author Response · Author response to Decision Letter 1]

10 Jun 2021

Response to Reviewers.

Dear reviewers’ kindly fine in this later the response to the questions posted.

The average annual temperature of the study area is 19.30c. During the risk factor analysis of the data, the demographic, socioeconomic and nutritional characteristics were all included together in the multivariate analysis. It was done after a bivariate analysis were performed. 

The investigator captured the aspect of clinical manifestation by using a questionnaire tool. These questions were asked to the participants to consult them of the history of clinical manifestation event related to that of the illness.

---

## [Decision Letter · Decision Letter 2]

13 Jul 2021

PONE-D-20-04369R2

Investigating the risk factors for seroprevalence and the correlation between CD4+ T-cell count and humoral antibody responses to Toxoplasma gondii infection amongst HIV patients in the Bamenda Health District, Cameroon.

PLOS ONE

Dear Dr. Enah,

Thank you for submitting your manuscript to PLOS ONE. After careful consideration, we feel that it has merit but does not fully meet PLOS ONE’s publication criteria as it currently stands. Therefore, we invite you to submit a revised version of the manuscript that addresses the points raised during the review process.

There are few grammatical/typo errors that need to be addressed before proceed to the next step as suggested by the reviewers.

We look forward to receiving your revised manuscript.

Kind regards,

Mohammed Abdelfatah Mosa Alhoot, PhD

Academic Editor

PLOS ONE

Journal Requirements:

Reviewers' comments:

Reviewer's Responses to Questions

**Comments to the Author**

1. If the authors have adequately addressed your comments raised in a previous round of review and you feel that this manuscript is now acceptable for publication, you may indicate that here to bypass the “Comments to the Author” section, enter your conflict of interest statement in the “Confidential to Editor” section, and submit your "Accept" recommendation.

Reviewer #3: All comments have been addressed

Reviewer #4: All comments have been addressed

2. Is the manuscript technically sound, and do the data support the conclusions?

Reviewer #3: Yes

Reviewer #4: Yes

3. Has the statistical analysis been performed appropriately and rigorously? 

Reviewer #3: Yes

Reviewer #4: Yes

4. Have the authors made all data underlying the findings in their manuscript fully available?

Reviewer #3: Yes

Reviewer #4: Yes

5. Is the manuscript presented in an intelligible fashion and written in standard English?

Reviewer #3: Yes

Reviewer #4: Yes

6. Review Comments to the Author

Reviewer #3: Comments to authors:

The manuscript was improved. However, minor edits should be made.

Lines 69-71: please delete “It would be useful to present data regarding the seroprevalence and risk factors associated with the infection amongst human immunodeficiency virus (HIV) of T. gondii from different areas of the world.”

I suggest to replace with this statement:

“Knowledge of T. gondii seroprevalence and its associated risk factors is important to predict the risk of infection in humans.”

Reviewer #4: The authors have adequately addressed comments raised in previous review. This time only 2 corrections in text (typing errors)

7. PLOS authors have the option to publish the peer review history of their article (what does this mean?). If published, this will include your full peer review and any attached files.

Reviewer #3: No

Reviewer #4: No

---

## [Author Response · Author response to Decision Letter 2]

14 Aug 2021

Responses to Review Comments for Manuscript: PONE-D-20-04369R2

Thank you for pointing out the corrections that should be done on our manuscript. We have dealt with them as recommended/advised; all the points raised have been addressed; grammatical and typographical errors inclusive. We have equally removed reference 24; this is because it has not been cited anywhere in the manuscript. We have also highlighted the changes introduced. We hope we have answered all the questions and concerns satisfactorily and therefore look forward to your anticipated action. 

REVIEWER 3

Comment: Please delete ‘’ It would be useful to present data regarding the seroprevalence and risk factors associated with the infection amongst human immunodeficiency virus (HIV) of T. gondii from different areas of the world.

Response: The statement has been deleted and replaced with‘’Knowledge of T. gondii seroprevalence and its associated risk factors is important to predict the risk of infection in humans’’ as suggested. Please, see lines 71 – 72 on the revised manuscript with tract changes

Raymond B. NYASSA, PhD

Corresponding author

---

## [Editor Report · Decision Letter 3]

20 Aug 2021

Investigating the risk factors for seroprevalence and the correlation between CD4+ T-cell count and humoral antibody responses to Toxoplasma gondii infection amongst HIV patients in the Bamenda Health District, Cameroon.

PONE-D-20-04369R3

Dear Dr. Enah,

We’re pleased to inform you that your manuscript has been judged scientifically suitable for publication and will be formally accepted for publication once it meets all outstanding technical requirements.

Kind regards,

Mohammed Abdelfatah Mosa Alhoot, PhD

Academic Editor

PLOS ONE
---

## [Editor Report · Acceptance letter]

23 Nov 2021

PONE-D-20-04369R3 

Investigating the risk factors for seroprevalence and the correlation between CD4+ T-cell count and humoral antibody responses to *Toxoplasma gondii* infection amongst HIV patients in the Bamenda Health District, Cameroon. 

Dear Dr. Nyasa:

I'm pleased to inform you that your manuscript has been deemed suitable for publication in PLOS ONE. Congratulations! Your manuscript is now with our production department. 

Kind regards, 

on behalf of

Dr. Mohammed Abdelfatah Mosa Alhoot 

Academic Editor

PLOS ONE